# Genetic and Genomic Diversity in a Tarwi (*Lupinus mutabilis* Sweet) Germplasm Collection and Adaptability to Mediterranean Climate Conditions

**Norberto Guilengue [1], Sofia Alves [1], Pedro Talhinhas [1,2,*] and João Neves-Martins [1]**

[1]   Instituto Superior de Agronomia, Universidade de Lisboa, 1349-017 Lisbon, Portugal;
      guilenguen@gmail.com (N.G.); sofiaalves@isa.ulisboa.pt (S.A.); nevesmartins@isa.ulisboa.pt (J.N.-M.)
[2]   LEAF, Linking Landscape, Environment, Agriculture and Food, Instituto Superior de Agronomia,
      Universidade de Lisboa, 1349-017 Lisbon, Portugal
*   Correspondence: ptalhinhas@isa.ulisboa.pt; Tel.: +351-2-1365-3249

**Abstract:** *Lupinus mutabilis* (tarwi) is a species of Andean origin with high protein and oil content and regarded as a potential crop in Europe. The success in the introduction of this crop depends in part on in depth knowledge of the intra-specific genetic variability of the collections, enabling the establishment of breeding and conservation programs. In this study, we used morphological traits, Inter-Simple Sequence Repeat markers and genome size to assess genetic and genomic diversity of 23 tarwi accessions under Mediterranean conditions. Phenotypic analyses and yield component studies point out accession LM268 as that achieving the highest seed production, producing large seeds and efficiently using primary branches as an important component of total yield, similar to the *L. albus* cultivars used as controls. By contrast, accession JKI-L295 presents high yield concentrated on the main stem, suggesting a semi-determinate development pattern. Genetic and genomic analyses revealed important levels of diversity, however not relatable to phenotypic diversity, reflecting the recent domestication of this crop. This is the first study of genome size diversity within *L. mutabilis*, revealing an average size of 2.05 pg/2C (2001 Mbp) with 9.2% variation (1897–2003 Mbp), prompting further studies for the exploitation of this diversity.

**Keywords:** *Lupinus mutabilis*; genetic diversity; morphological traits; ISSR; genome size; Mediterranean climate

## 1. Introduction

The genus *Lupinus* includes more than 280 species [1], approximately 90% of which are native and widely distributed throughout the American continent [2,3], with greater inter- and intra-specific genetic variability than in Euro-African species. *Lupinus mutabilis* Sweet (also known as tarwi, chocho, altramuz and Andean lupin) is native from the Andean region in South America. The species is auto- and allogamic with wide variability of flower, stem and seed colours, and exhibits indeterminate growth [4,5]. It has been domesticated in the Andean region and used for grain production, forage, green manure, fixing atmospheric nitrogen and soil conservation [4,6]. In spite of their high alkaloid content [7,8], tarwi seeds have high nutritional value, containing up to 53% protein and 24% lipids [9]. The nutritional attributes found in tarwi are supposedly better than those in soybeans [10] and for this reason it is called Andean soybean [11]. Tarwi protein is rich in globulins (43%–45%) and albumins (8%–9%) and the oil has high quality and does not require industrial removal of the linolenic acid like in soybean [12,13]. Additionally, low alkaloid (<0.1%) lines have been selected in *L. mutabilis* [14].

Tarwi exhibits key traits of domestication, including indehiscent pods and seeds with permeable tegument, representing a locally important crop in several Andean areas [15]. Recently the species

*L. piurensis* was considered the wild relative from which tarwi would have evolved until arriving at the domesticated form known nowadays [16]. According to this hypothesis, no wild specimens of *L. mutabilis* exist and the species would have suffered a classic domestication bottleneck no later than 2600 years before present time [16], leading to a recognizably low genetic diversity of tarwi [17]. Nevertheless, the crop conceals important morphological variability, which is related to the high variability of agroecological conditions across its native range [18]. For instance, small plants occur in the Potosi region, where the altitude exceeds 3500 m and of low temperatures and precipitation prevail. Branched and tall plants are found in the Andean valleys of Bolivia and Southern Peru with more than 50% of their production centred on the main stem. Highly branched plants with over 1.8 m in height, with long vegetative period and little production in the main stem occur in Colombia, Ecuador and northern Peru, under frost-free climates [18].

Due to its high plasticity, tarwi has a wide adaptation to varied soils, precipitation and temperature regimes [6]. In the light of this broad adaptation, attempts have been made in order to introduce tarwi to European conditions [12] to reduce local dependence on imported soybeans. As such, seeds harvested in the Andean region have been used during several years to select plants with determined growth in the Mediterranean conditions. As a result, a germplasm collection was created focused on promising accessions. The success in introducing this species in this region will depend in part on the deep knowledge of the genetic variability of this collection. Thus, understanding the genetic variability is extremely important for the establishment of future breeding and conservation programmes [19].

Research on crop genetic variability has been based on morphological descriptors and molecular markers as the main tools [20–23]. The morphological descriptors are used to generate relevant information about the description and classification of the germplasm collections in order to allow efficient use in breeding programmes [24,25]. Morphological analysis and molecular markers can be used together to generate more reliable and consistent information. Contrary to morphological descriptors, molecular markers have the advantage to not depend on the environment, phenotype and stage of development of the plant [26]. Several DNA markers are available and can be used in genetic diversity studies, among which are Inter-Simple Sequence Repeat (ISSR) markers. ISSR markers allow preliminary screening of germplasm collections and have been used to perform genetic mapping, phylogenetic and evolutionary studies because of their good repeatability, high polymorphism, easy handling and low cost [27–30]. ISSR analyses thus enable the selection of contrasting accessions that, together with pertaining morphological traits, can be selected for further characterisation using more informative markers, such as Simple Sequence Repeats. In addition to the use of molecular markers, in recent years the use of nuclear DNA content information to explain intra-specific genetic diversity has been increasing [31–33]. The DNA content is important for understanding molecular, cellular and evolutionary genomic mechanisms [34]. Flow cytometry is widely used for DNA content estimation due to its simplicity and efficacy [35]. This technique has been applied successfully in the estimation of nuclear DNA content in different species. In particular, flow cytometry was employed to differentiate *Lupinus* species based on the genome size [36]. However, there are few studies addressing intraspecific variability in *L. mutabilis* based on morphological traits, ISSR [37,38] and on DNA content. This prompts a need to characterise *L. mutabilis* germplasm collections in depth, both under genotypic and phenotypic perspectives. Instituto Superior de Agronomia (ISA), Portugal, has one of the most important collections of *Lupinus* in the world, containing over 1300 *Lupinus* accessions, including *L. mutabilis*. However, little is known about the genetic variability in this collection. The present study aims to evaluate genetic and genomic diversity in 23 *L. mutabilis* accessions present in ISA collection using 37 morphological traits, six ISSR markers and genome size data, contributing simultaneously to assess its adaptability to Mediterranean climate conditions and to provide genotypic data.

## 2. Materials and Methods

### 2.1. Plant Materials

A total of 23 *L. mutabilis* accessions were selected from the ISA *Lupinus* germplasm including five accessions provided by the Julius Kühn-Institut (JKI), Germany (e.g., Table 2). *Lupinus albus* cultivars Misak and Mihai were used as reference in the morphological characterization because of their high adaptation to the Mediterranean conditions and as outgroups/standards in the ISSR marker and genome size analyses.

### 2.2. Morphological Analysis

Field experiments were conducted at Tapada de Ajuda in Lisbon (coordin: 38.709133, −9.182976, alt: 60 m) on a vertisol in the 2016/17 (sowing date: 29 December) and 2017/18 (sowing date: 18 December) seasons under rain-fed conditions. Meteorological data were collected daily from the weather station located adjacent to the field. Soil water balances were calculated according to Allen et al. [39].

The experimental design adopted was randomized block with three replicates. Each replicate was composed of 26 1.8 $m^2$-plots with 20 plants in each plot (immediately surrounded by a 60 cm-wide edge of *L. albus* 'Misak' plants to avoid border effects) and the total number of plots in the assay was 78. For morphological characterization, 10 plants of each plot were selected as recommended by Talhinhas et al. [40].

Data of morphological characterization were obtained based on *Lupinus* spp. descriptors [41], as listed in Table 1. Yield components and vegetative traits were analysed considering a two-factor experimental design (genotype and year), with differences being statistically analysed using the Kruskal Wallis test. Characteristics for multivariate analysis were selected based on correlation coefficients and heritability values [40]. Variables with correlation above 0.85 were considered redundant and thus one of them was excluded. Meanwhile, variables with low heritability (<65%) were also excluded, as these were explained by environmental factors.

Univariate analysis (UA) was performed to compare each individual characteristic across the accessions. Before running the UA, normality and homogeneity of variances was tested. Since data did not follow normal distribution and the variance was not homogeneous, an analysis of variance (ANOVA) based on rank transformation for non-parametric analysis was performed [42]. Post-hoc Tukey's honest significant difference (HSD) test of means was performed for all variables at 5% significance. Afterwards, broad sense heritability ($H^2$), genotypic variance ($\sigma^2{}_g$), phenotypic variance ($\sigma^2{}_P$), phenotypic coefficient of variance (PCV) and genotypic coefficient of variance (GCV) were estimated to understand the genetic variation between accessions and environment, as well as the genetic effects on different traits, following Mazid et al. [43].

Multivariate analysis was performed for all 25 accessions and all characteristics selected and represented in a single graphic, as described by Talhinhas et al. [40]. Standardized morphological data transformation (mean = 0, and standard deviation = 1) was performed before conducting multivariate analysis. Cluster analysis was performed based on Euclidean distance and average method for the 25 accessions. A dendrogram was constructed using an unweighted pair group method of arithmetic mean (UPGMA) algorithm. Principal component analysis (PCA) was performed and eigenvectors and eigenvalues were projected to visualize the components. All analyses were performed in the RStudio program version 1.1.456 (The R consortium, Boston, USA).

**Table 1.** List of morphological traits evaluated in the experiment, method and unit of measurement.

| Acronym | Trait [1] | Method (unit) |
|---|---|---|
| DUF | Days from sowing until flowering [3] | Counting (nr [4]) |
| NLMS | Number of leaves on the main stem | Counting (nr) |
| HUFF | Height up to first flower | Metric meas. [5] (cm) |
| ADNL | Average distance between leaves | = HUFF/NLMS (cm) |
| NPMS | Number of pods on the main stem | Counting |
| PLMS | Pod length on the main stem | Metric meas. (cm) |
| PWMS | Pod width on the main stem | Metric meas. (cm) |
| RBLWPMS | Ratio length-width of pods on the main stem | = PLMS/PWMS (dim. [6]) |
| NSMS | Number of seeds on the main stem | Counting (nr) |
| SLMS | Seed length on the main stem [2] | Metric meas. (cm) |
| SWMS | Seed width on the main stem | Metric meas. (cm) |
| RBLWSMS | Ratio length-width of seed on the main stem [2,3] | = SLMS/SWMS (dim.) |
| NSPMS | Number of seed per pod on the main stem | Counting (nr) |
| WSMS | Weight of seeds on the main stem | Weighting (g) |
| TSWMS | Thousand seeds weight on the main stem [2] | = WSMS/NSMS*1000 (g) |
| NPB | Number of primary branches | Counting (nr) |
| ADBPB | Average distance between primary branches [2,3] | = HUFF/NPB (cm) |
| SLPB | Sum of the length of primary branches | Metric meas. (cm) |
| ALPB | Average length of primary branches | = SLPB/NPB (cm) |
| PBL | Proportion of leaves with branches [2,3] | = NPB/NLMS (%) |
| NPPB | Number of pods on primary branches [2] | Counting (nr) |
| NSPB | Number of seeds on primary branches [2] | Counting (nr) |
| NSPPB | Number of seeds per pod on primary branches | = NSPB/NPPB (nr) |
| NPPPB | Number of pods per primary branch | = NPPB/NPB (nr) |
| WSPB | Weight of seeds on primary branches [2] | Weighting (g) |
| WSPPB | Thousand seeds weight per primary branches | Weighting (g) |
| TSWPB | Thousand seeds weight on primary branches | = WSPB/NSPB × 1000 (g) |
| TBL | Total branch length | = SLPB+ HUFF |
| TNP | Total number of pods [2] | Counting (nr) |
| TNS | Total number of seeds | Counting (nr) |
| TNSPP | Total number of seeds per pod | = TNS/TNP (nr) |
| TW | Total seed weight | Weighing (g) |
| PSMS | Percentage of seed weight on the main stem | = WSMS/TW (%) |
| PSPB | Percentage of seed weight on primary branches [3] | = WSPB/TW (%) |
| TTSW | Total thousand seeds weight | = TW/TNS × 1000 (g) |
| SWBLR | Seed weight/total branch length ratio | = TW/TBL × 100 (g/m) |

[1] Characteristics related with secondary and tertiary branches were excluded due to insufficiency of data; [2] Redundant or non-independent characteristics excluded of multivariate analysis based on the correlation coefficient ($r > 0.85$); [3] Characteristics excluded of multivariate analysis due to presenting low value of heritability (<0.65); [4] number; [5] Metric measurement; [6] dim.—dimensionless.

## 2.3. Molecular Analysis

Young but fully expanded leaves of the 23 *L. mutabilis* accessions and of the two *L. albus* reference cultivars were collected and immediately frozen in liquid nitrogen and stored at −80 °C. Freeze-dried vegetal material was used for DNA extraction using the DNeasy® Plant mini kit (Qiagen, Hilden, Germany) according to the manufacturer instructions. The DNA quality and quantity were estimated using spectrophotometry in the Gen5 program, and electrophoresis using a 1% agarose gel. The stock solution of DNA was diluted with sterilized water to make a working solution with a concentration of 10 ng/μL to be used in amplifications.

For molecular characterization, six ISSR primers were selected (Table 6) from those reported by Talhinhas et al. [44] based on preliminary analyses of a limited set of accessions. The Polymerase Chain Reaction (PCR) amplification for all primers was carried out under the following conditions: pre-denaturation 4 min at 94 °C, 40 cycles of 30 s at 94 °C, 45 s at 52 °C and 2 min at 72 °C, and a final extension at 72 °C for 10 min. The PCR reactions were performed in a final volume of 10 μL

containing 20 ng of DNA, 0.5 μM of primer and 5 μL of dNTP + *Taq* DNA polymerase (NZYTaq II DNA polymerase, NZYTech, Lisbon, Portugal). After amplification, products were separated by electrophoresis in a 2% agarose gel stained using GreenSafe Premium (NZYTech).

The ISSR bands were scored in a binary matrix as presence (1) or absence (0) for each accession and for each fragment size. Based on the binary matrix, parameters such as percentage of polymorphic and monomorphic bands were determined and discriminatory power of primers was calculated based on the polymorphic information content (PIC), effective multiplex ratio (EMR), resolving power (RP) and marker index (MI). PIC value is the probability for detecting polymorphism by a primer or primers combination between two randomly drawn genotypes and can be calculated using the formula $PIC = 1 - \Sigma pi^2$, where pi is the frequency of occurrence of polymorphic bands in different primers [45]. The effective multiplex ratio was calculated using the formula $EMR = np\beta$; where ß is the fraction of polymorphic markers and is estimated after considering the number of polymorphic loci (np) and non-polymorphic loci (nnp) as ß = np/(np + nnp) [46]. Marker index (MI) is the primer capacity to detect polymorphic loci among different genotypes and was calculated as *EMRxPIC*. Resolving power (RP) is the ability of primers to distinguish between genotypes and was calculated as $RP = \Sigma Ib$, where Ib is the informative fragments and can take values of: $1 - [2|0.5 - p|]$; $p$ is the proportion of total genotypes containing the band [47]. Genetic similarity was obtained according to the Jaccard similarity index. The results were used for the construction of ISSR and morphological traits dendrograms, in order to evaluate the similarity relations between the genotypes. Dendrograms were constructed based on UPGMA grouping and the ISSR results were correlated with morphological traits.

### 2.4. Flow Cytometry

For each accession, young leaves in healthy conditions were randomly collected and immediately analysed in the laboratory. Nuclear DNA content was measured by flow cytometry. *Solanum lycopersicum* 'Stupické' (2C = 1.96 pg; [48]) was tested as DNA standard but its genome size showed to be too close to that of *L. mutabilis.* Therefore, we tested *L. albus* as DNA standard (2C = 1.20 pg; [49]) and for such *L. albus* 'Misak' was validated as standard by comparison to *S. lycopersicum* 'Stupické' and *Raphanus sativus* 'Saxa' (2C = 1.11 pg; [48]). Each *L. mutabilis* accession, together with the standard, was chopped with a razor blade in the presence of 1 mL of buffer (Woody Plant Buffer; [50]). The nuclear suspension obtained was then separated from plant debris using a 30 μm nylon filter. After filtration, 50 μg/mL of propidium iodide (PI; Sigma-Aldrich) were added to stain DNA and 50 μg/mL of RNase (Sigma-Aldrich) ware added to prevent staining of double stranded RNA. The samples were maintaining at room temperature and analyzed using a CyFlow Space flow cytometer (Sysmex, Norderstedt, Germany) equipped with a 30 mW green solid-state laser emitting at 532 nm for optimal PI excitation. The reproducibility of results were assessed using five independent replicates for each accession. FloMax software v2.4d (Sysmex) was used to measure nuclear DNA content and three graphics were generated from data measurement: fluorescence pulse integral in linear scale (FL); fluorescence pulse integral in linear scale versus time; and fluorescence pulse integral in linear scale versus side light scatter in logarithmic scale (SSC). The absolute DNA amount of a sample was calculated based on the values of the G1 peak means, as suggested by Doležel and Bartoš [51]:

$$Sample\ 2C\ DNA\ content = \frac{Sample\ G1\ peak\ mean}{standard\ G1\ peak\ mean} \times Standard\ 2C\ DNA\ Content \qquad (1)$$

The results generated from 2C DNA (in picogram) were transformed to million base pairs using the following conversion: 1 pg = 978 Mbp [52]. Coefficient of variation (CV, %) of G1 peaks in the FL histograms, and estimates of the CV of the genome size of each accession were used to assess the reliability of the results. Intra-specific genome size comparison was carried out using Kruskal Wallis test ($\alpha$ = 0.05) because genome size data did not exhibit normal distribution. Data analysis was done in RStudio Program Version 1.1.456.

## 3. Results

### 3.1. Morphological Characterization and Genetic Parameters among Accessions

Studies on the genetic variability are important because they generate relevant data for breeding programmes and can be used as basis for development and selection of superior genotypes. Here, we used morphological characterization and genetic parameters to evaluate the variability of a *L. mutabilis* germplasm collection under Mediterranean conditions.

Meteorological conditions during the trial were typical of the Mediterranean climate (Figure S1), although rainfall was well below average during autumn and winter and above average during spring in 2017/18, while rain was scarce in April 2016. Two-way ANOVA based on rank transformation was performed and revealed that all morphological traits exhibited significant difference at $p$-value < 0.05. Tables 2 and 3 show the mean values, homogeneous groups and $p$ value obtained in a two-factor experimental design for morphological and reproductive characteristics, respectively. Differences analysis results for morphological traits of each year are given in Supplementary Material (Table S1). The statistical analysis of the results depict those genotypes showing differences that were consistent over the two years.

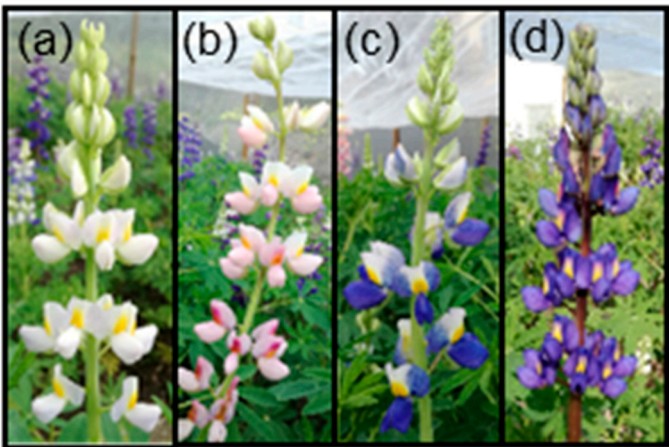

**Figure 1.** *Lupinus mutabilis* flowers and stem colours: (**a**) green stem, white wings, white standard with yellow central spot; (**b**) green stem, pale pink wings (more intense as flower matures), pale pink standard with yellow central spot (central spot turning dark pink as the flower matures); (**c**) green stem, blue wings, standard blue in the marginal area, white in the intermediate and yellow in the central spot; (**d**) purple stem, purple wings, purple standard with yellow central spot.

The results presented in Table 2 reveal that the average number of days from sowing to flowering (DUF) ranged between 80.8 (accession JKI-L309) and 103.4 (accession XM1-39) for *L. mutabilis*, spanning 23 days, while for *L. albus* cultivars the average was 98.1 days. The number of leaves on the main stem (NLMS) ranged between 12.8 and 18.0, while the number of primary branches (NPB) ranged from 2.6 to 4.6. The average numbers of leaves (NLMS) and branches (NPB) were 15.9 and 3.4, respectively, showing similar values to *L. albus*. On average, 21% of main stem leaves axillae harboured primary branches (PBL), with a minimum of 16% for accession JKI-L295 and a maximum of 26% for accession LM231. The average height (HUFF) of *L. mutabilis* plants was 55.7 cm, ranging between 40.0 cm (JKI-L309) and 68.9 cm (Inti), while the average for *L. albus* cultivars was 40.4 cm. In both experiments the accession JKI-L309 grew less than other accessions. The total length of primary branches (SLPB) varied between 81.9 cm for accession JKI-L309 and 163.1 cm for accession LM231, with an average of 122.8 cm. The total stem length (main stem and primary branches, TBL) attained a global average of 179.4 cm (164.7 for *L. albus*), varying between 122.0 cm (accession JKI-L309) and 229.4 cm (LM231). The JKI-L309 accession presented low values for TBL, suggesting that this may be a semi-determinate

genotype. Stem and flower colors varied among accessions, with no clear correlation to morphologic traits. Figure 1 depicts the four groups of flower and stem colors.

**Table 2.** Average values (and homogeneous groups [1]) for vegetative traits of 23 *Lupinus mutabilis* accessions and two *L. albus* cultivars ('Mihai' and 'Misak'), obtained upon analysis of variance (ANOVA) based on rank transformation.

| Accession | SFC [2] | DUF [3] | NLMS | HUFF | NPB | SLPB | PBL | TBL |
|---|---|---|---|---|---|---|---|---|
| JKI-L309 | D | 80.8 a | 12.9 a | 40.0 a | 2.8 abc | 81.9 a | 21.6 efgh | 122.0 a |
| JKI-L377 | B | 92.8 b | 12.8 a | 47.7 bc | 2.7 ab | 87.2 a | 19.9 cde | 137.7 ab |
| MUTAL | A | 93.5 b | 14.8 b | 60.0 gh | 3.3 efg | 152.0 h | 22.8 efgh | 213.1 i |
| JKI-L210 | C | 96.4 c | 15.8 def | 50.0 bcd | 3.0 cde | 89.1 a | 19.1 bc | 142.2 bc |
| LM13 | C | 97.7 cd | 16.5 fg | 53.0 cdef | 3.9 ijk | 132.0 cdefg | 23.3 efghi | 185.6 efgh |
| LM81 | C | 97.8 cd | 15.2 bcd | 54.3 cdef | 3.3 cdef | 127.0 bcdef | 21.7 defgh | 182.4 defgh |
| JKI-L295 | A | 98.6 cd | 16.2 f | 60.7 fgh | 2.6 a | 114.1 b | 16.3 a | 175.2 def |
| LM268 | A | 98.8 cd | 16.3 f | 60.9 gh | 3.6 hij | 133.3 defgh | 22.2 efgh | 194.2 fghi |
| PRT79 | C | 98.8 cd | 17.4 hi | 65.1 hi | 3.5 hij | 136.2 efgh | 20.4 cde | 202.3 ghi |
| I82 | D | 99.9 cd | 15.9 def | 52.4 cde | 3.4 fghi | 115.8 bc | 21.2 def | 169.0 de |
| LM34 | D | 100.1 cd | 15.2 bcd | 50.9 bcd | 3.7 hijk | 124.5 bcde | 24.3 i | 175.6 de |
| Potosi-ALE | C | 100.3 cd | 15.0 b | 53.6 cde | 3.5 fghi | 134.2 bcdef | 23.4 fghi | 188.7 efgh |
| LM27 | A | 100.4 cd | 15.9 cdef | 55.8 efgh | 3.5 fghi | 125.3 bcdef | 21.9 defgh | 181.7 efgh |
| Potosi-ISA | C | 100.5 cd | 16.7 fg | 58.9 efgh | 3.4 fghi | 118.5 bcd | 20.7 cde | 177.5 def |
| LM18 | C | 100.5 cd | 17.7 ghi | 54.6 defg | 3.7 hij | 127.8 bcdef | 20.8 cde | 182.5 defg |
| CM157 | C | 100.6 cd | 15.4 bcde | 49.1 bcd | 3.0 bcd | 111.4 b | 18.6 bc | 160.7 cd |
| XM-5 | B | 101.2 cd | 16.4 fg | 58.5 efgh | 3.5 fghi | 125.0 bcdef | 21.3 defg | 184.1 efgh |
| LM32 | C | 101.6 d | 16.2 ef | 53.7 cdef | 3.4 fgh | 125.3 bcdef | 21.5 def | 179.6 defg |
| LM231 | C | 102.4 d | 18.0 i | 64.8 hi | 4.6 k | 163.2 gh | 25.5 ghi | 229.4 hi |
| P20993 | B | 102.6 d | 15.1 bc | 55.3 defg | 3.6 ghij | 125.1 bcdef | 23.5 hi | 180.4 defg |
| INTI | C | 102.6 cd | 17.7 i | 68.9 i | 3.3 def | 147.0 fgh | 18.6 b | 218.7 i |
| SBP | C | 103.0 d | 16.0 f | 58.6 efgh | 3.2 def | 119.7 bcdef | 19.7 bcd | 179.4 defgh |
| XM1-39 | C | 103.4 d | 16.2 f | 54.6 defg | 3.1 def | 110.3 bc | 18.7 bc | 166.4 de |
| Mihai | | 97.6 cd | 16.6 fgh | 42.6 ab | 3.7 hijk | 122.1 bcdef | 22.5 defgh | 164.7 cde |
| Misak | | 98.6 cd | 18.8 j | 38.2 a | 3.8 jk | 120.7 bcdefg | 20.4 cde | 158.9 cd |
| *p*-value [4] | | 0.000 | 0.000 | 0.000 | 0.000 | 0.000 | 0.000 | 0.000 |
| *p*-value [5] | | 0.000 | 0.000 | 0.000 | 0.000 | 0.000 | 0.020 | 0.000 |
| *p*-value [6] | | 0.000 | 0.000 | 0.000 | 0.000 | 0.000 | 0.000 | 0.000 |

[1] Homogeneous groups—accessions sharing the same letter for each trait are not statistically different; [2] SFC—stem and flower colour, according to Figure 1; [3] Full name of acronyms and description of the respective morphologic traits are given in Table 1; [4] *p*-value taking into account the accessions; [5] *p*-value taking into account the experiments; [6] *p*-value taking into account the interaction between accessions and experiments.

The total seed weight (TW) per plant (Table 3) varied 3.7×, ranging between 3.7 g per plant (accession JKI-L210) and 13.8 g per plant (accession LM268), the latter attaining a projected productivity estimated at 1533 kg/ha, although less than half of the total yield of the *L. albus* accessions. Dissecting yield components evidences additional variability among the accessions (Table 3). The total number of pods (TNP) per plant varied nearly 2.1×, with a maximum of 25 pods per plant for accession Potosi-ISA. The total number of seeds (TNS) reached a maximum of 67.9 seeds per plant (accession LM34). The average number of seeds per pod (TNSPP) is 2.7, ranging between 2.2 (accession LM268) and 3.3 (accession JKI-L210). The total thousand seeds weight (TTSW) attained a global average of 187.0 g, varying 2.85× between 101.4 g (accession JKI-L210) and 289.2 g (accession LM268). LM268 was the only *L. mutabilis* accession producing more yield on the primary branches than on the main stem (40% and 60% of total yield on the main stem and on primary branches, respectively), following a similar pattern to that of *L. albus* cultivars. The accessions JKI-L295 and JKI-L210 produced about 80% of seed weight on the main stem. Unlike accession LM268, several *L. mutabilis* accessions reached superior seed yields (over 10 g per plant) while concentrating over 60% of their yield on the main stem: CM157, I82 and LM27. For the comparison of seed yield and vegetative development (Table 3), the seed weight/total branch length ratio (SWBLR) was calculated. SWBLR average was 2.0 g of seeds per

meter of branch length in *L. mutabilis* (23.1 g/m in *L. albus*), ranging between 1.1 g/m (accession LM32) and 3.3 g/m (accession Mutal).

**Table 3.** Average values (and homogeneous groups [1]) for yield components of 23 *Lupinus mutabilis* accessions and two *L. albus* cultivars ('Mihai' and 'Misak'), obtained upon ANOVA based on rank transformation.

| Accession | TNP [2] | TNS | TNSPP | TTSW | PSMS | PSPB | TW | SWBLR |
|---|---|---|---|---|---|---|---|---|
| JKI-L210 | 11.5 a | 36.6 ab | 3.3 fg | 101.4 a | 87.8 | 12.2 | 3.7 a | 1.3 a |
| INTI | 13.7 a | 37.4 ab | 2.7 bcd | 154.0 cd | 74.6 | 25.4 | 5.7 b | 2.3 ef |
| XM1-39 | 13.9 ab | 36.1 ab | 2.8 bcd | 180.4 jkl | 78.1 | 21.9 | 6.3 b | 1.3 a |
| JKI-L377 | 19.8 cdef | 61.3 cde | 3.0 ef | 103.7 a | 72.2 | 27.8 | 6.3 b | 3 def |
| JKI-L309 | 16.1 bc | 49.2 cd | 3.1 efg | 159.7 ef | 72.2 | 27.8 | 7.8 c | 3.1 fg |
| SBP | 16.8 cd | 43.6 bc | 2.6 bc | 188.7 lm | 77.0 | 23.0 | 8.0 cd | 1.7 abc |
| LM32 | 22.1 efg | 56.7 def | 2.7 bcd | 150.8 cde | 67.3 | 32.7 | 8.4 cde | 1.1 ab |
| Potosi-ALE | 25.0 gh | 62.2 efg | 2.6 b | 182.7 bc | 59.1 | 40.9 | 8.4 cdef | 1.5 abcd |
| LM34 | 24.9 gh | 67.9 fg | 2.7 bcde | 131.6 b | 55.0 | 45.0 | 8.5 cde | 1.6 abcd |
| JKI-L295 | 18.0 cde | 48.0 cd | 2.7 bcd | 188.4 m | 80.3 | 19.7 | 9.0 def | 2.2 bcdef |
| LM231 | 22.9 fgh | 59.9 def | 2.6 bcd | 169.4 fgh | 56.2 | 43.8 | 9.5 cdef | 1.5 abcd |
| MUTAL | 19.2 def | 52.2 cd | 2.8 bcd | 184.7 lm | 67.2 | 32.8 | 9.6 def | 3.3 g |
| LM81 | 23.1 gh | 63.1 fg | 2.9 bcde | 157.8 ef | 57.7 | 42.3 | 9.7 fghi | 1.8 abcde |
| LM13 | 22 fgh | 57.5 def | 2.7 bcd | 179.0 ijk | 70.3 | 29.7 | 9.9 efgh | 1.8 abcde |
| XM-5 | 23.8 gh | 59.7 def | 2.6 bc | 170.1 ghi | 61.2 | 38.8 | 9.9 defg | 1.9 abcde |
| CM157 | 23.0 gh | 64.9 efg | 2.8 cde | 157.3 def | 61.6 | 38.4 | 10.0 fghij | 2.1 abcdef |
| LM27 | 23.0 efg | 62.0 def | 2.8 bcde | 178.4 ghijk | 64.1 | 35.9 | 10.5 fghij | 1.8 abcde |
| PRT79 | 22.5 gh | 62.8 fg | 2.8 de | 177.6 hijk | 58.3 | 41.7 | 11.1 ghijk | 2.1 abcde |
| P20993 | 24.8 hi | 67.7 g | 2.8 bcde | 168.9 fg | 58.0 | 42.0 | 11.1 ijk | 2.1 abcdef |
| I82 | 21.8 efg | 64.5 fg | 3.1 efg | 175.4 ghij | 62.5 | 37.5 | 11.2 ghijk | 2.2 def |
| Potosi-ISA | 23.7 gh | 64.3 fg | 3.1 bcde | 184.0 klm | 55.7 | 44.3 | 11.5 jk | 2.3 bcdef |
| LM18 | 23.2 gh | 66.1 efg | 2.9 def | 190.5 lm | 52.1 | 47.9 | 12.6 hijk | 2.2 cdef |
| LM268 | 22.1 gh | 50.5 cd | 2.2 a | 289.2 n | 40.9 | 59.1 | 13.8 k | 2.7 abcdef |
| Mihai | 25.7 hi | 101.2 h | 4.0 h | 370.7 n | 28.8 | 71.1 | 36.7 l | 23.3 abcde |
| Misak | 28.8 i | 98.0 h | 3.4 g | 381.5 n | 45.0 | 54.9 | 37.2 l | 22.9 abcde |
| *p*-value [3] | 0.000 | 0.000 | 0.000 | 0.000 | 0.000 | 0.000 | 0.000 | 0.000 |
| *p*-value [4] | 0.040 | 0.000 | 0.000 | 0.000 | 0.969 | 0.756 | 0.004 | 0.000 |
| *p*-value [5] | 0.000 | 0.000 | 0.002 | 0.000 | 0.000 | 0.000 | 0.000 | 0.000 |

[1] Homogeneous groups—accessions sharing the same letter for each trait are not statistically different; [2] Full name of acronyms and description of the respective morphologic traits are given in Table 1; [3] *p*-value taking into account the accessions; [4] *p*-value taking into account the experiments; [5] *p*-value taking into account the interaction between accessions and experiments.

In Table 4 are presented the average, phenotypic and genotypic variance with their respective coefficient of variation and heritability for the 2016/17 and 2017/18 experiments. Higher values of phenotypic and genotypic variances were observed for TSWPB, TSWMS, TBL, TNS, SLPB, NSPB and WSPPB (see Table 1 for definitions). Conversely, low values were observed for ADNL, RBWLPMS, RBWLSMS, NSPMS, NPB, TNSPP, PSMS, PBL, PSPB, PWMS and SLMS. The highest phenotypic and genotypic coefficients of variation were obtained in 2017/18 for NPPB (33.64% and 29.18%) and NSPB (37.68% and 30.89%), while low values were observed in both years for DUF, RBLWSMS and RBLWPMS. In general, values of the phenotypic coefficient of variation were relatively higher than genotypic. For all characteristics, heritability ranged from 0% to 100%. Most of the characteristics exhibited high heritability, while lower and medium values were found for NSPMS (0), RBWLSMS (0.53), RBWLPMS (0.64, 0.45), NSPPB (0.42, 0.39), TNSPP (0.52), DUF (0.49), SWMS (0.57), PSPB and PBL (0.63). These characteristics exhibiting low and medium values of heritability were excluded for the multivariate analysis.

**Table 4.** Genetic parameters estimated for 38 quantitative traits among 23 *Lupinus mutabilis* accessions.

| Traits | Average | | Phenotypic Variance | | Genotypic Variance | | PCV [1] | | GCV [2] | | H [2,3] | |
|---|---|---|---|---|---|---|---|---|---|---|---|---|
| | 2016/17 | 2017/18 | 2016/17 | 2017/18 | 2016/17 | 2017/18 | 2016/17 | 2017/18 | 2016/17 | 2017/18 | 2016/17 | 2017/18 |
| TSWMS | 184.15 | 10.94 | 1368.99 | 1779.1 | 1368.82 | 1599.4 | 20.09 | 21.91 | 20.09 | 20.77 | 1 | 0.9 |
| TSWPB | 156 | 3.04 | 1423.64 | 1562.07 | 1423.39 | 1466.89 | 24.19 | 24.64 | 24.18 | 23.88 | 1 | 0.94 |
| TTSW | 158.29 | 177.98 | 1145.83 | 1561.49 | 1113.07 | 1339.14 | 21.39 | 22.20 | 21.08 | 20.56 | 0.97 | 0.86 |
| DUF | 87.67 | 3.04 | 17.21 | 23.25 | 16.59 | 10.85 | 4.73 | 4.52 | 4.65 | 3.09 | 0.96 | 0.47 |
| WSPPB | 52.29 | 23.32 | 148.79 | 382.43 | 140.77 | 289.57 | 23.33 | 30.14 | 22.69 | 26.22 | 0.95 | 0.76 |
| HUFF | 73.1 | 31.24 | 83.85 | 26.45 | 78.7 | 23.64 | 12.53 | 12.29 | 12.14 | 11.61 | 0.94 | 0.89 |
| NLMS | 15.21 | 23.32 | 2.14 | 1.36 | 2 | 1.22 | 9.62 | 7.09 | 9.29 | 6.73 | 0.93 | 0.9 |
| SLMS | 0.95 | 51.2 | 0 | 0.6 | 0 | 0.57 | 6.72 | 8.19 | 6.48 | 7.97 | 0.93 | 0.95 |
| PWMS | 1.57 | 7.57 | 0.01 | 0.01 | 0.01 | 0.01 | 7.12 | 8 | 6.75 | 7.83 | 0.9 | 0.96 |
| SWBLR | 4.31 | 192.52 | 1.31 | 0 | 1.17 | 0 | 26.54 | 20.07 | 25.09 | 18.33 | 0.89 | 0.83 |
| TBL | 220.4 | 352.93 | 1183.56 | 421 | 1044.3 | 344.87 | 15.61 | 14.55 | 14.66 | 13.17 | 0.88 | 0.82 |
| ALPB | 46.86 | 10.94 | 17.46 | 27.53 | 15.11 | 22.04 | 8.92 | 14.9 | 8.29 | 13.33 | 0.87 | 0.8 |
| TW | 9.49 | 31.24 | 7.7 | 6391.49 | 6.73 | 6049.35 | 29.24 | 22.65 | 27.33 | 22.04 | 0.87 | 0.95 |
| ADNL | 4.84 | 10.48 | 0.16 | 0 | 0.14 | 0 | 8.35 | 9.57 | 7.72 | 8.82 | 0.85 | 0.85 |
| PSMS | 0.66 | 99 | 0.01 | 0.02 | 0.01 | 0.01 | 16.93 | 19.84 | 15.5 | 17.91 | 0.84 | 0.81 |
| PSPB | 0.34 | 0.07 | 0.01 | 0.01 | 0.01 | 0.01 | 32.29 | 26.56 | 29.56 | 21.14 | 0.84 | 0.63 |
| SLPB | 147.29 | 2.64 | 796.89 | 291.04 | 671.56 | 218.18 | 19.17 | 17.23 | 17.59 | 14.92 | 0.84 | 0.75 |
| NSMS | 31.29 | 0.66 | 48.96 | 38.56 | 40.63 | 25.37 | 22.36 | 19.88 | 20.37 | 16.12 | 0.83 | 0.66 |
| WSMS | 5.69 | 5.54 | 1.82 | 2.76 | 1.52 | 2.3 | 23.74 | 29.96 | 21.66 | 27.39 | 0.83 | 0.84 |
| WSPB | 4.19 | 3.6 | 2.68 | 2.72 | 2.19 | 2.19 | 39.08 | 45.87 | 35.36 | 41.15 | 0.82 | 0.8 |
| ADBPB | 25.11 | 3.56 | 12.34 | 14.96 | 9.59 | 10.17 | 13.99 | 22.22 | 12.33 | 18.32 | 0.78 | 0.68 |
| NPMS | 9.67 | 7.94 | 3.85 | 2.44 | 2.98 | 1.88 | 20.29 | 14.9 | 17.84 | 13.09 | 0.77 | 0.77 |
| NPB | 3.12 | 3.03 | 0.17 | 0.23 | 0.13 | 0.19 | 13.18 | 13.41 | 11.52 | 12.25 | 0.76 | 0.84 |
| NPPPB | 3.11 | 1.39 | 0.67 | 0.81 | 0.51 | 0.52 | 26.38 | 29.55 | 22.9 | 23.77 | 0.75 | 0.65 |
| PBL | 0.21 | 1.27 | 0 | 0 | 0 | 0 | 11.77 | 17.57 | 10.18 | 13.96 | 0.75 | 0.63 |
| TNS | 62.05 | 3.56 | 191.54 | 140.03 | 142.42 | 104.68 | 22.3 | 23.11 | 19.23 | 19.98 | 0.74 | 0.75 |
| NSPB | 26.83 | 0.41 | 72.78 | 77.2 | 52.66 | 51.89 | 31.8 | 37.68 | 27.05 | 30.89 | 0.72 | 0.67 |
| NPPB | 10.09 | 0.18 | 8.17 | 13.53 | 5.77 | 10.18 | 28.32 | 33.64 | 23.8 | 29.18 | 0.71 | 0.75 |
| TNP | 21.62 | 64.89 | 22.19 | 20.2 | 15.86 | 15.97 | 21.79 | 22.64 | 18.42 | 20.13 | 0.71 | 0.79 |
| PLMS | 6.96 | 9.46 | 0.25 | 7.88 | 0.17 | 7.87 | 7.17 | 29.04 | 5.93 | 29.65 | 0.68 | 1 |
| RBLWSMS | 1.28 | 19.85 | 0 | 0 | 0 | 0 | 2.66 | 4.81 | 2.13 | 0 | 0.64 | 0.45 |
| SWMS | 0.75 | 160.41 | 0 | 0.38 | 0 | 0.36 | 7.29 | 8.1 | 5.48 | 7.88 | 0.57 | 0.95 |
| RBLWPMS | 4.46 | 141.02 | 0.06 | 0.09 | 0.03 | 0.08 | 5.6 | 6.61 | 4.06 | 6.13 | 0.53 | 0.86 |
| TNSPP | 2.98 | 10.48 | 0.07 | 0.09 | 0.04 | 0.08 | 9.1 | 11.59 | 6.55 | 10.5 | 0.52 | 0.82 |
| NSPPB | 2.74 | 4.64 | 0.18 | 6.04 | 0.08 | 2.37 | 15.5 | 30.97 | 10.04 | 19.41 | 0.42 | 0.39 |
| NSPMS | 3.48 | 1.52 | 0.21 | 0.2 | 0 | 0.14 | 13.28 | 14.87 | 0.67 | 12.17 | 0 | 0.67 |

[1] Phenotypic coefficient of variation; [2] Genotypic coefficient of variation; [3] Broad sense heritability.

　　Correlation is an important test and is used to assess relationship and associations between variables and is frequently applied in several studies. The results generated after correlation coefficient analysis using the Spearman method for morphological traits in experiments of 2016/17 and 2017/18 are presented in Tables S2-1 and S2-2. Results show positive correlation for most traits. Total weight (TW) was positively correlated ($p \leq 0.05$) with 13 variables in 2016/17: SLPB (0.6**), TBL (0.69**), NPB (0.60**), PBL (0.47**), TNP (0.57**), NPPB (0.65**), WSMS (0.70**), TSWMS (0.59**), NSPB (0.57**), WSPB (0.91**), TWSPB (0.56**), NPPPB (0.59) and TNS (0.65**). In 2017/18 TW was positively correlated with eleven variables: TNS (0.80**), TSWMS (0.68**), NPB (0.56**), SLPB (0.61**), NPPB (0.70**), NSPB (0.72**), NPPPB (0.62**), TSWPB (0.55**), TNP (0.72**), TBL (0.60**), WSMS (0.73**) and WSPB (0.92**). Also, positive correlation was found between TW and NSPPB (0.44**), SLMS (0.76**), SWMS (0.66**), SWBLR (0.84**), NSPPB (0.55**), PSPB (0.61**) and PLMS (0.53**). Significant ($p \leq 0.05$) and positive correlation was also reported between TNS and NSPMS, SLPB, NPPB, NSPB, NPPPB, NSPPB, TNP, ADNL, PBL, WSMS, TSWMS, NPB, SWPB, TW, SWBLR, TBL, PSPB and WSPB in both years of experiments.

　　From the correlation data, heat maps were constructed (Figures 2 and 3) using euclidean distances and the UPGMA method, where in the vertical columns are the clusters of morphological traits while in the horizontal lines are the clusters of accessions. Dark red colors represent lower values while the dark blue are higher values. Figure 2 corresponds to the heat map obtained from 2016/17 data and Figure 3 was obtained with 2017/18 data. Six groups of morphological traits could be drawn in Figure 2 and five in Figure 3. In both figures, group 1 is related to the reproductive capacity of pods, defined in the Figure 2 by the characteristics NSMS, SWBLR and TW and in Figure 3 by PLMS, WSMS, TNS, TW,

SWBLR, NSPPB and NPPB. In Figure 2, group 2 is composed by characteristics related to the distance between leaves and primary branches in the main stem (ADNL and ADBPB), while group 2 in Figure 3 includes vegetative and reproductive traits related to the main stem: average distance between leaves (ADNL), number of seeds per pod on the main stem (NSPMS) and ratio between length and width of pod on the main stem (RBLWPMS). Only one characteristic (PSMS) defines group 3 in both figures. In Figure 2, group 4 includes characteristics related with total number of seeds (TNS), proportion of leaves with branches (PBL) and percentage of seed on the primary branches (PSPB). In Figure 3, group 4 includes vegetative and reproductive traits related to the main stem (NSMS, NPMS, NPB and NLMS). Group 5 (Figure 2), includes traits related to the reproductive capacity on the main stem (TSWMS, TSWPB, PWMS and PLMS). The same group on Figure 3 is related with pod and seed size (thousand seeds weight, pod and seed size parameters) and traits that include primary branches (ALPB, SLPB and WSPPB). Group 6 is mostly defined by vegetative characteristics (TBL, ALPB, HUFF and DUF).

Cluster I represent three JKI accessions in both figures, which is discriminated by morphological groups 5 and 6 (four reproductive and five vegetative characteristics) in Figure 2. In Figure 3 Cluster I is discriminated by morphological groups 1 (seven reproductive characteristics), 4 (with three reproductive and two vegetative characteristics) and 5 (five reproductive and two vegetative characteristics). Cluster II (Figure 2) is composed by 6 accessions defined by groups 1 and 4 (five reproductive and one vegetative characteristics) and, in Figure 3, by 19 accessions and does not exhibit a defined pattern. In both figures, Cluster III with accession LM268 only, is characterized by high values in groups 1 and 5 (SWBLR, TW, PWMS, TSWPB and TSWMS) in Figure 2 and high values in group 5 and low values in groups 2–4 in Figure 3. Cluster IV in Figure 2 does not exhibit a defined pattern. For Figure 3, Cluster IV is represented by the *L. albus* cultivars and is characterized by high values for all heat map in most characteristics. This cluster is defined by three groups of morphological traits: group 1 (seven reproductive variables), group 2 (three reproductive and one vegetative characteristics) and group 5 (five reproductive and two vegetative characteristics).

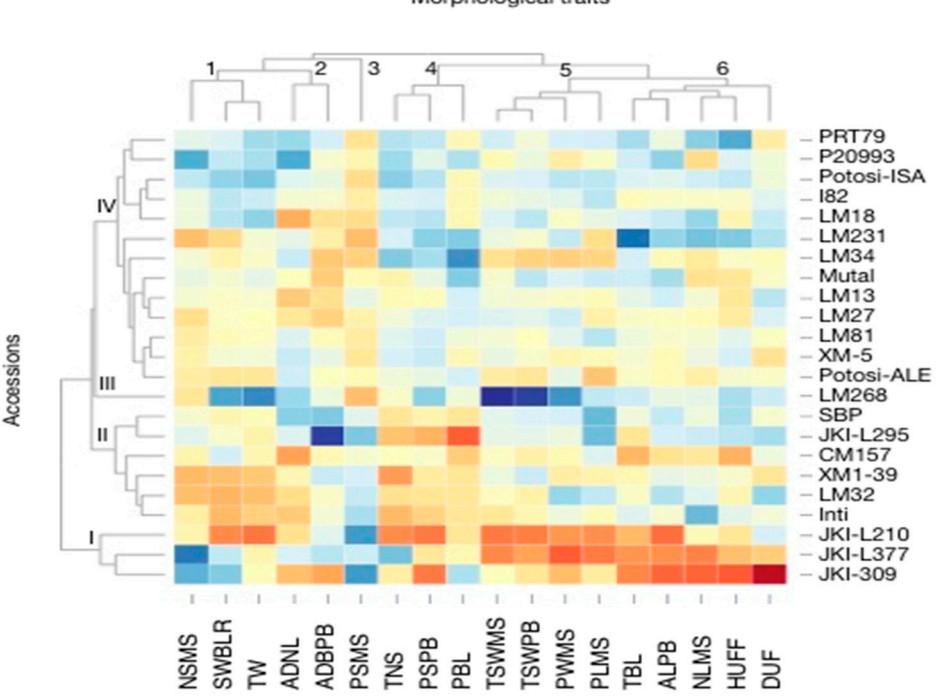

**Figure 2.** Heat map of the 23 accessions of *L. mutabilis* obtained from morphological characterization data for the 18 traits, where red and blue boxes indicate low values and high values respectively.

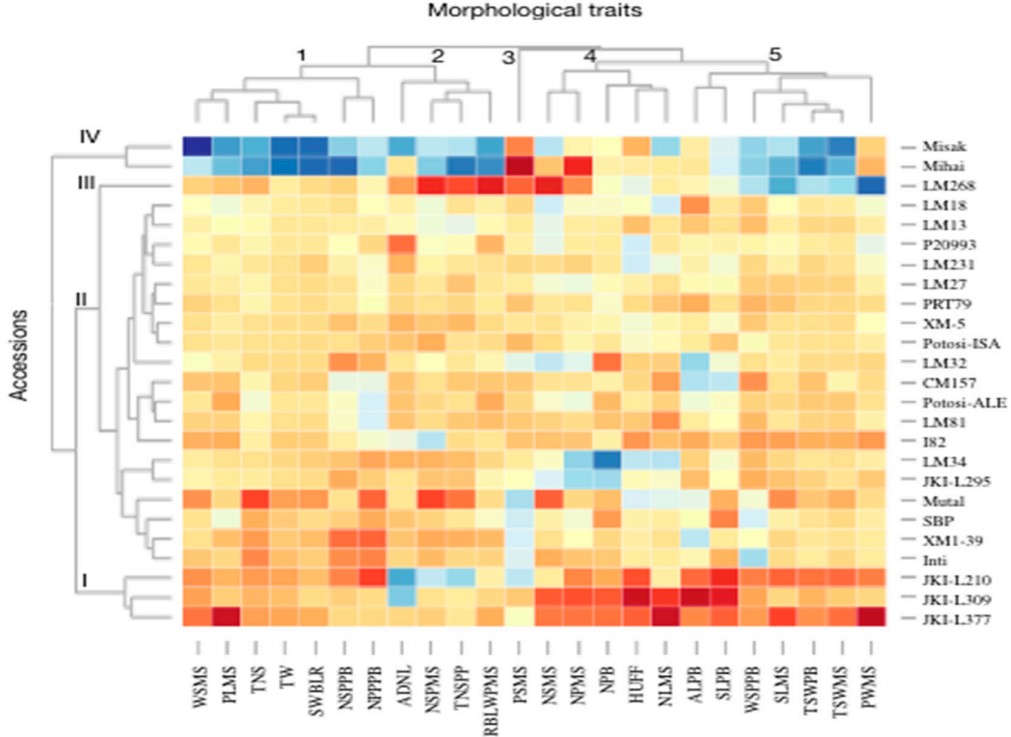

**Figure 3.** Heat map of the 25 accessions obtained from morphological characterization data for the 24 traits, where red and blue boxes indicate low values and high values respectively.

Principal component analysis (PCA) confirmed the cluster analysis results (Figures 2 and 3). For instance: cluster I is localized oppositely for many vectors of groups 5 and 6 (TSWMS, TSWPB, PWMS PLMS, TBL, ALPB, NLMS, HUFF, and DUF) for Figure 4 and a similar scenario can be observed in Figure 5 were the vectors defined by groups 2 and 4 (HUFF, PWMS, ALPB, NPB, NLMS, SLPB, WSPPB, SLMS, TSWMS, TSWPB, WSMS and PLMS) are in the opposite position, thus justifying the low values of these characteristics in those groups (Figures 4 and 5). In both figures the LM268 accession presents high values for many characteristics among *L. mutabilis* accessions. Cluster IV (Figure 5) is composed of two accessions that present high values for 17 vectors (characteristics marked in cluster analysis with blue color). In this cluster eight vectors (RBLWPMS, TSWMS, TSWPB, NSPPB, SWBLR, TW, WSMS and TNSPP) are highlighted by presenting the highest scores, with TW being the longest vector. The first three PCs projected in the biplot (Figure 4) show a clear separation of the four cluster and all together account for 75.1% of the total variation. The first component accounts for 40.3% of variation, with PWMS, TBL, PSPB, PSMS and ALPB accounting heavily for this variation. The second PC accounts for 21.4% of the variation, with TNS, NSMS and TW being the most important variables. The third PC explains another 13.4% of the variation, with the most important variables being PBL, SWBLR and PLMS. In Figure 5 the first three component explain 76.4% of total variation. For the first principal component, characteristics TW, SWBLR, TSWMS, TSWPB and TNS contribute more, explaining 43.9% of total variation. HUFF, ADNL, PWMS, TNSPP, and NSPMS are most important variables for second component; this component accounts for explanation 20.4% of variation, while NSMS, NPMSM LMS and PSMS account for 12% of variation in the third component.

### 3.2. Diversity Assessed by Molecular Markers

The six selected ISSR primers used for analysis of 23 accessions resulted in the production of 37 reproducible bands (Table 5 and Figure 6). Of those, 11 (29.7%) bands were polymorphic and the remaining 26 (70.3%) were monomorphic. The total number of bands per primer ranged between four (GT$_8$YC) and eight (HVH(TG)$_7$), while the percentage of polymorphic bands per primer ranged from 0

to 50%. The average for each primer was 6.2 bands. Polymorphism information content (PIC), which is used in genetics as a measure of polymorphism for a marker locus, ranged from 0.23 (HVH(TG)$_7$) to 0.72 (AG$_8$YT). Effective multiplex ratio (EMR) had its minimum value with AG$_8$YC (0) and maximum in GT$_8$YC (2.25). The resolving power (RP) parameter used to detect the differences between a large number of genotypes ranged from 5.48 (AG$_8$YT) to 13.58 (HVH(TG)$_7$). The minimum and maximum values for marker index were registered for AG$_8$YC (0) and GT$_8$YC (0.54) primers, respectively.

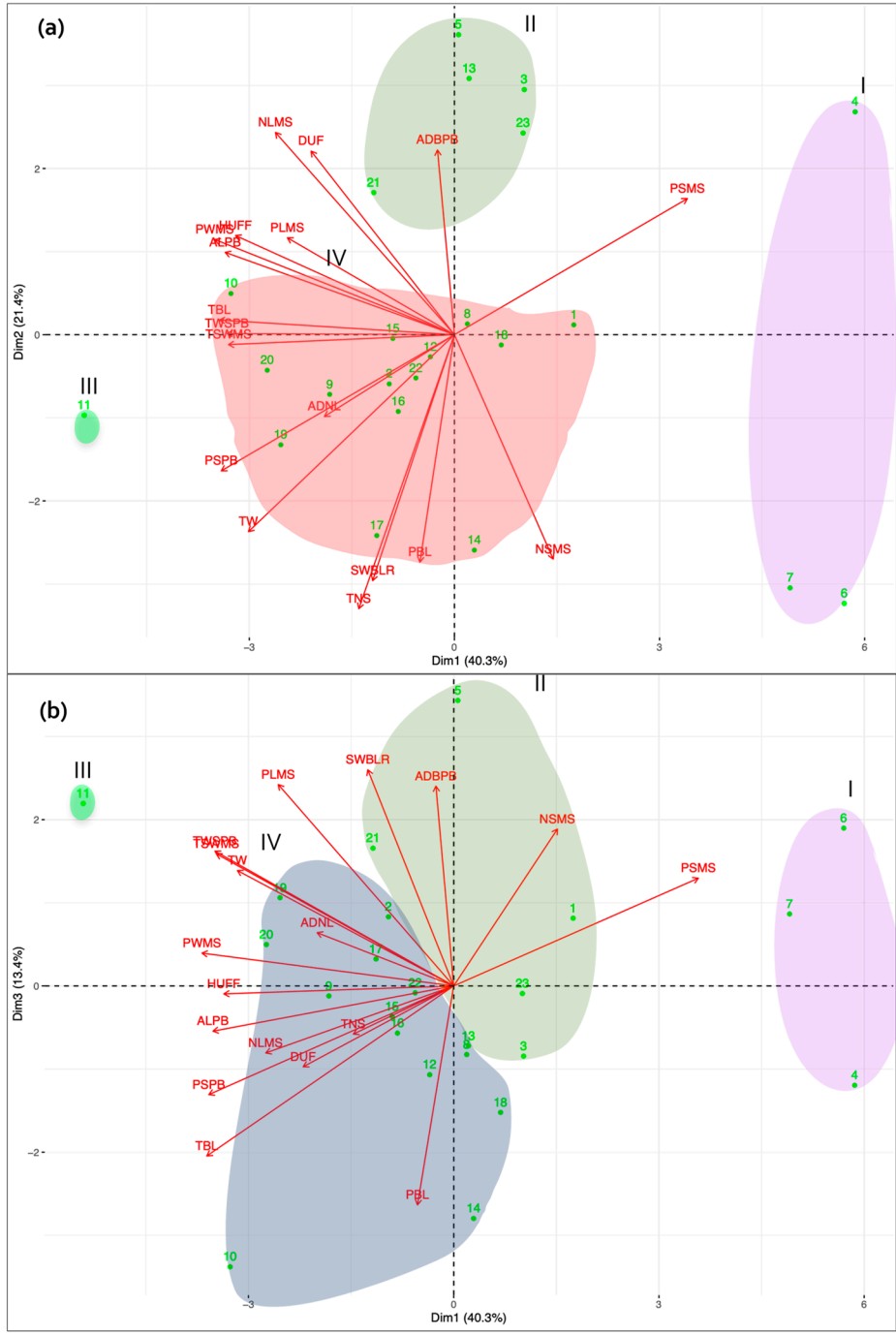

**Figure 4.** Representation in two dimensions (first and second dimensions—(panel **a**)—of principal component analysis explain 61.7% of the variability, while the inclusion of the third dimension—(panel **b**)—raises the three-dimensional space to explain 75.1% of the variability) of normalized original data of morphological characterization of the 23 *Lupinus mutabilis* accessions in a space defined by the vectors and own values.

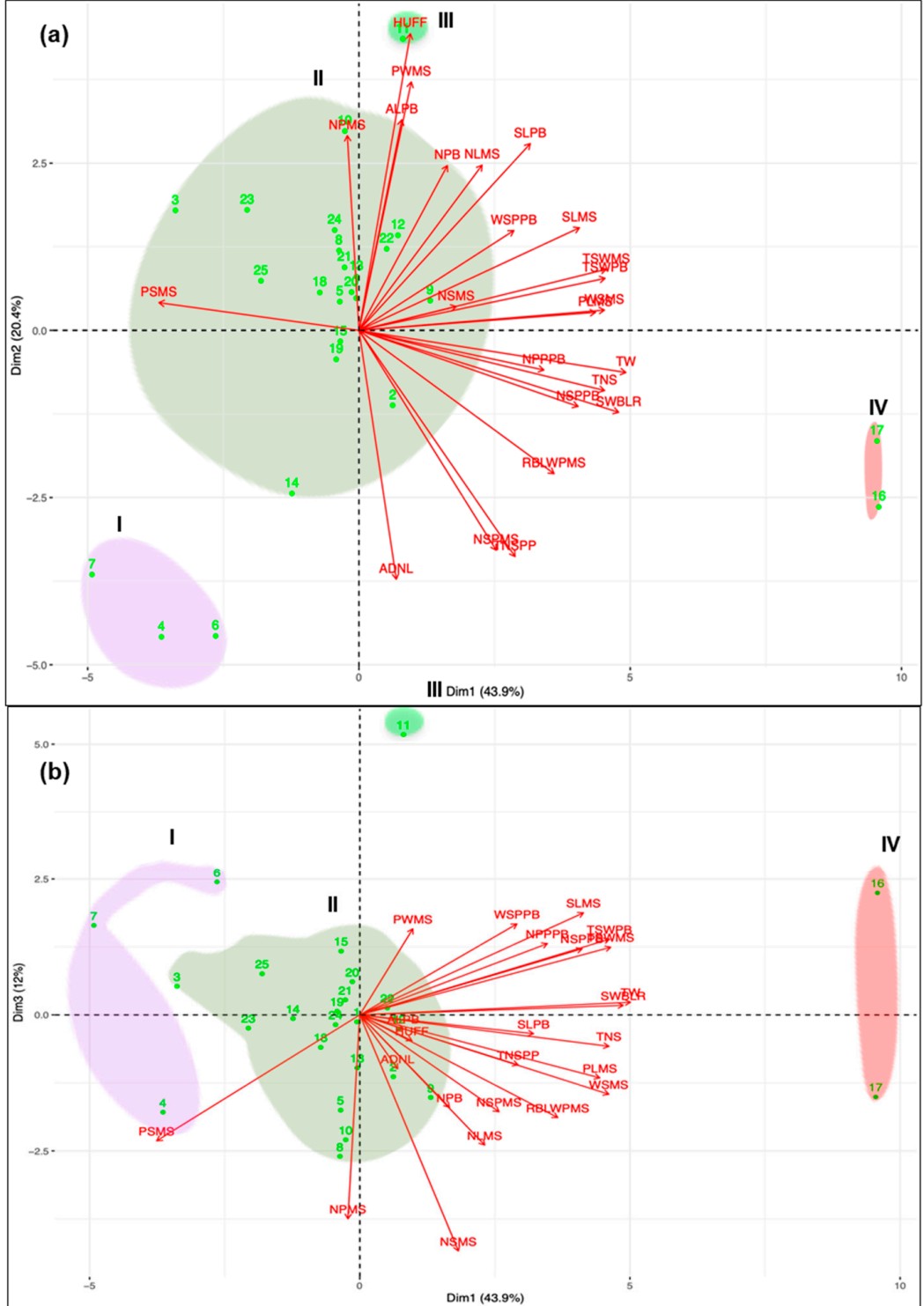

**Figure 5.** Representation in two dimensions (first and second dimensions—(panel **a**)—of principal component analysis explain 64.3% of the variability, while the inclusion of the third dimension—(panel **b**)—raises the three-dimensional space to explain 76.3% of the variability) of normalized original data of morphological characterization of the 23 *Lupinus mutabilis* accessions in a space defined by the vectors and own values. Numbers 1–25 encode accessions as detailed in Figure 4 (16 and 17 denote *L. albus* 'Mihai' and 'Misak' respectively).

**Table 5.** List of Inter-Simple Sequence Repeat (ISSR) primers used in this study, their total numbers of band per primer, polymorphic and monomorphic band and polymorphism percentage per primer.

| Primer | Bands | PB | MB | PB (%) | MB (%) | PIC | EMR | RP | MI |
|---|---|---|---|---|---|---|---|---|---|
| HVH(TG)$_7$ | 8 | 4 | 4 | 50 | 50 | 0.23 | 2 | 13.58 | 0.46 |
| GA$_8$YT | 6 | 1 | 5 | 16.66 | 83.33 | 0.71 | 0.16 | 5 | 0.11 |
| AG$_8$YT | 6 | 1 | 5 | 16.66 | 83.33 | 0.72 | 0.16 | 4.58 | 0.12 |
| GT$_8$YC | 4 | 3 | 1 | 75 | 25 | 0.24 | 2.25 | 6.42 | 0.54 |
| AG$_8$YC | 5 | 0 | 5 | 0 | 100 | 0.58 | 0 | 6 | 0.00 |
| AG$_8$YG | 8 | 2 | 6 | 25 | 75 | 0.48 | 0.5 | 10.33 | 0.24 |
| Total | 37 | 11 | 26 | | | | | | |
| Minimum | 4 | 0 | 1 | 0 | 25 | 0.23 | 0 | 4.58 | 0 |
| Maximum | 8 | 4 | 6 | 50 | 100 | 0.72 | 2.25 | 13.58 | 0.54 |
| Mean | 6.16 | 1.83 | 4.33 | 30.55 | 69.44 | 0.49 | 0.85 | 7.65 | 0.24 |

Notes: PB—polymorphic bands; MB—monomorphic bands; MB (%)—percentage of monomorphic bands; PB (%)—percentage of polymorphic bands; PIC—polymorphism information content; EMR—effective multiplex ratio; RP—resolving power; MI—marker index. The following primers: (CA)$_8$RY, (GA)$_8$YC, (GT)$_8$YC, (TCC)$_5$ and MR were included in the screening test but were rejected during selection. Eight of these primers were previous tested [53].

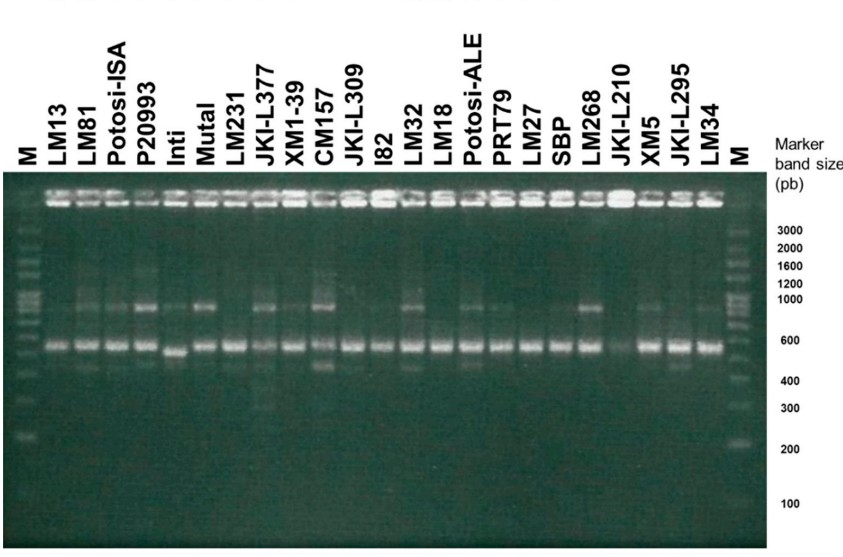

**Figure 6.** Example of ISSR amplification profiles for 23 *Lupinus mutabilis* accessions using the primer GA$_8$YT separated on a 2% agarose gel. M-NZYDNA Ladder VII marker. Numbers adjacent to accession names refer to coding used in Figure 5.

The similarity matrix was used to construct a dendrogram using the UPGMA method (Figure 7). The cophenetic correlation was 0.9058603, revealing little loss of information with transformation of similarity matrix to dendrogram. The dendrogram reveals five distinct groups. Cluster I is composed by 10 accessions of white, blue and pink flower colors and green and purple stem. Cluster II, containing 9 accessions, can be distinguished from the first group by the absence of purple stem and flower genotypes. Unlike cluster I and II, clusters III, IV and V are composed only by accessions that exhibit green stems and blue flowers. Cluster IV is represented by one accession and cluster III and V by two accessions each.

*3.3. Diversity Assessed by Genomic Traits*

*Lupinus albus* 'Misak' was validated as a DNA standard by comparison to *Solanum lycopersicum* 'Stupické' (Figure 8a,b) and *Raphanus sativus* 'Saxa' (data not shown) and estimated at 2C = 1.35 ± 0.0076 pg (1377.6 Mbp), with an average coefficient of variation of 3.47%.

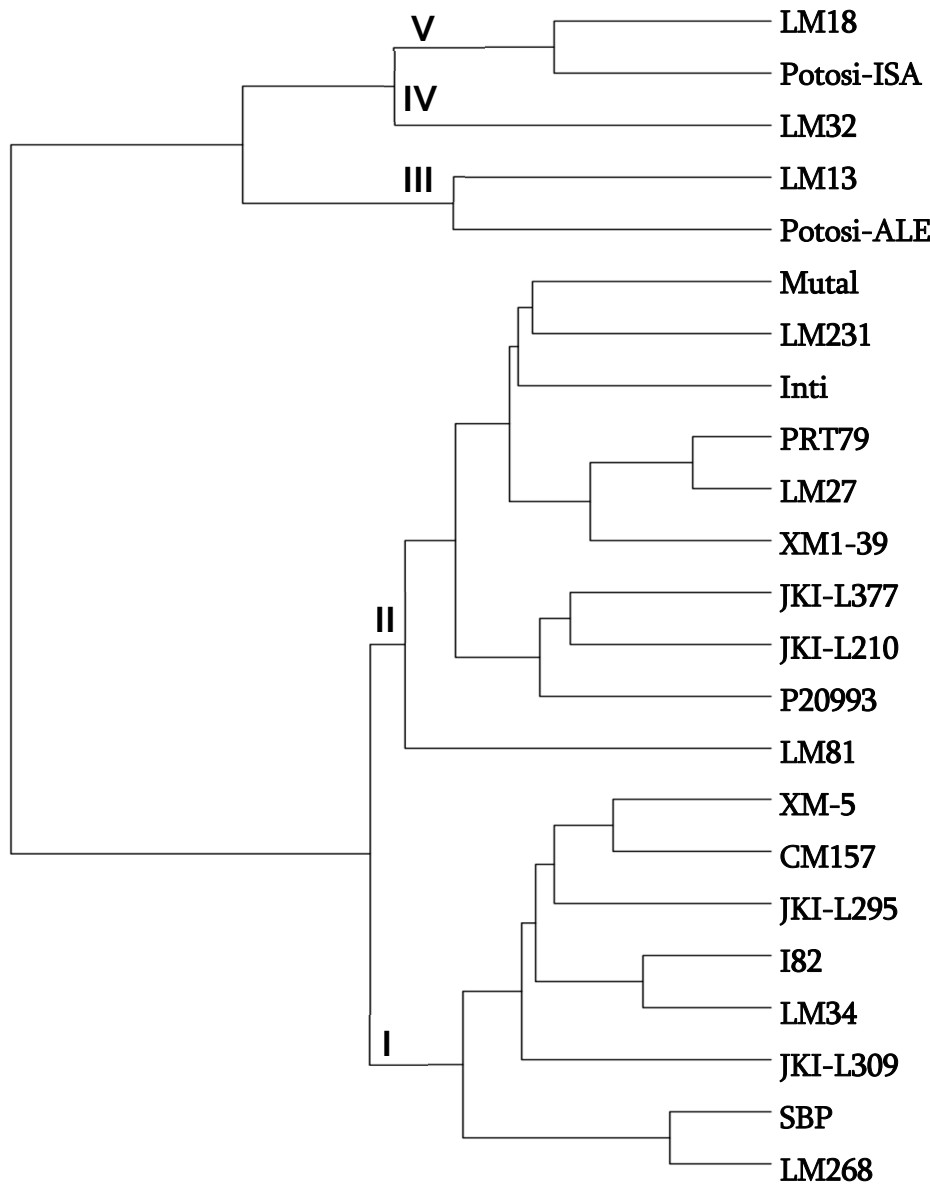

**Figure 7.** Dendrogram obtained by the unweighted pair group method of arithmetic mean (UPGMA) method from the coefficients of similarity (DICE) between the accessions of *Lupinus mutabilis* from six ISSR markers. *r* = 0. 9058603.

The *L. mutabilis* collection was thus analysed by comparison to *L. albus* 'Misak', as exemplified in Figure 8 (panels c,d). The average *L. mutabilis* genome size was estimated at 2C = 2.05 pg (2001.2 Mbp) with a 9.2% coefficient of variation, ranging from 1897.3 Mbp for accession SBP to 2083.2 Mbp for accession LM34 (Table 6). The results from a Kruskal–Wallis test performed for genome size reveal significant difference between accessions ($\chi^2$ = 94.845, *Df* = 23, *p* value = 0.000). No single accession showed to be statistically different from all the others, rather a continuum of accessions is depicted by the homogeneous groups produced (Table 6).

Genome size is an important criterion to study evolution at the intra-specific level, helping to understand conflicting pattern between morphological traits. In this study we evaluated the associations between genome size and morphological traits using Spearman correlation analysis for all 23 accessions for the two experiments. However, no single morphological trait presented strong correlation with genome size (Figure 9).

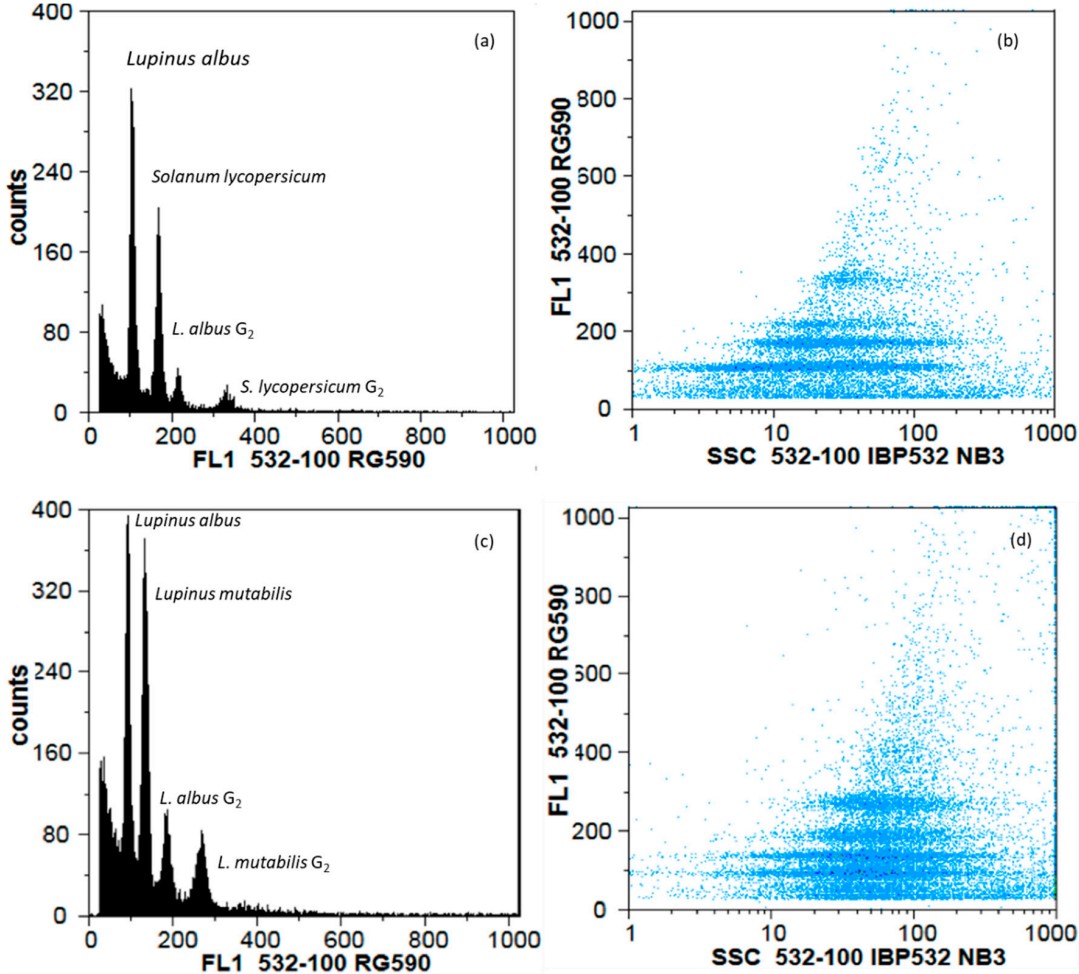

**Figure 8.** Flow cytometric analysis of relative fluorescence intensities (FL1) of propidium iodide-stained nuclei simultaneously isolated from: (**a**) and (**b**) *Solanum lycopersicum* 'Stupické' and *Lupinus albus*, for the validation of *L. albus* 'Misak' as DNA standard (2C = 1.35 pg); (**c**) and (**d**) *L. albus* 'Misak' and *L. mutabilis* accession LM231. (**a**) and (**c**) Histogram showing relative fluorescence intensities. (**b**) and (**d**) dot plots on side scatter (SSC) versus FL1.

**Table 6.** Genome size of the *Lupinus mutabilis* accessions estimated by flow cytometry.

| Accession | Genome Size (Mbp) | | H.G. [2] |
|---|---|---|---|
| | Average | StDev [1] | |
| **SBP** | 1897.3 | 18.4 | a |
| XM1-39 | 1907.3 | 17.0 | a |
| JKI-L378 | 1938.0 | 49.0 | ab |
| Prt-79 | 1957.4 | 16.2 | ab |
| JKI-L377 | 1961.4 | 16.3 | ab |
| Mutal | 1967.6 | 121.6 | abc |
| JKI-L295 | 1969.0 | 11.8 | abc |
| JKI-L210 | 1973.6 | 26.9 | bcd |
| LM13 | 1975.7 | 118.4 | bcd |
| JKI-L309 | 1979.7 | 37.6 | bcd |
| LM231 | 1984.1 | 100.4 | cd |
| LM18 | 1986.4 | 54.6 | cde |
| XM5 | 2009.1 | 20.1 | cde |
| P-20993 | 2021.5 | 22.8 | cde |
| Potosi-ISA | 2024.3 | 36.1 | cde |
| Potosi-ALE | 2024.8 | 19.3 | cde |

| Accession | Genome Size (Mbp) | | H.G. [2] |
| | Average | StDev [1] | |
|---|---|---|---|
| LM27 | 2027.7 | 23.9 | cde |
| LM32 | 2040.9 | 17.9 | de |
| CM157 | 2040.9 | 43.8 | de |
| I82 | 2041.6 | 10.9 | de |
| Inti | 2058.1 | 23.9 | ef |
| LM268 | 2078.9 | 10.1 | f |
| LM81 | 2080.2 | 13.1 | f |
| LM34 | 2083.2 | 17.3 | f |

[1] Standard deviation; [2] Homogeneous groups—accessions sharing the same letter are not statistically different.

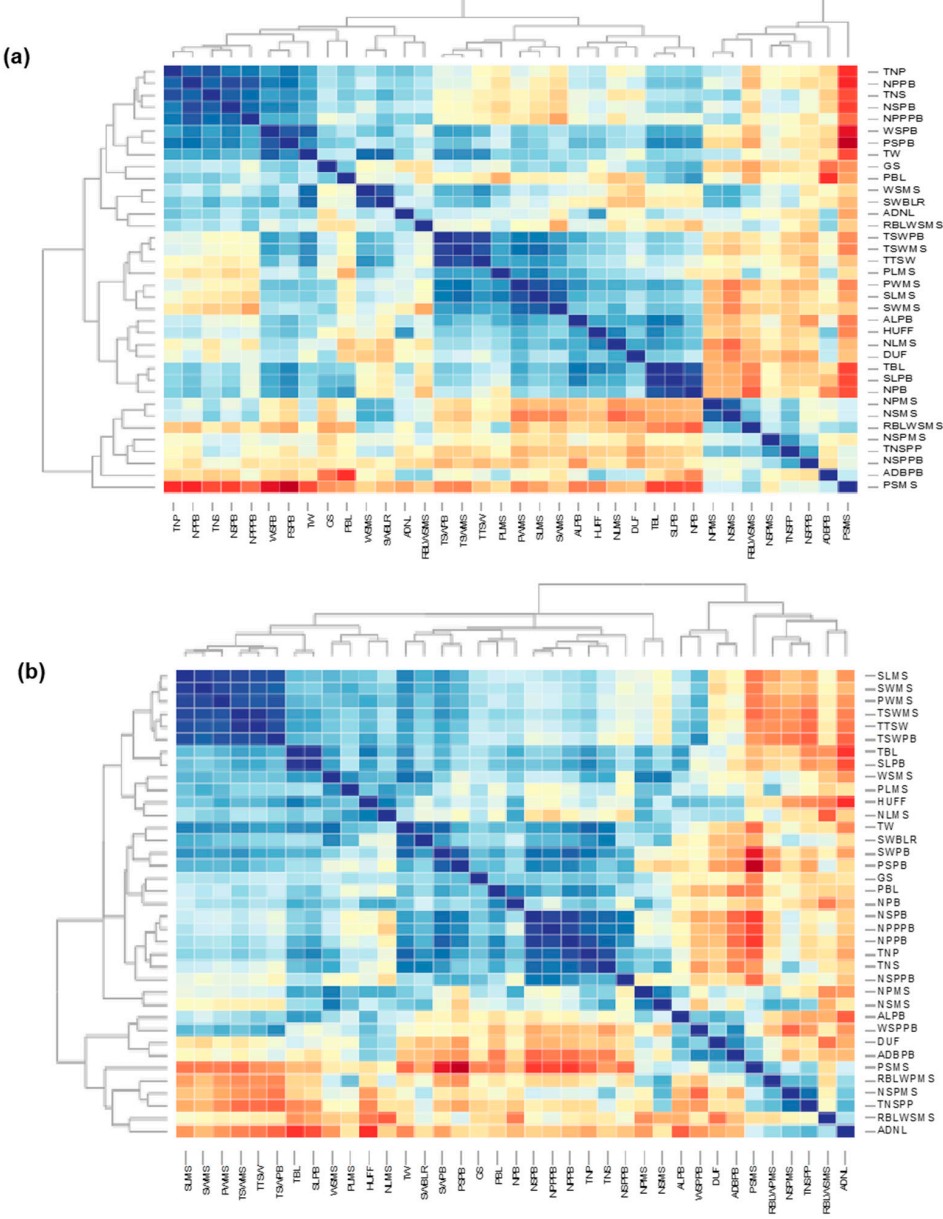

**Figure 9.** Heat map of the 23 *Lupinus mutabilis* accessions obtained from Spearman correlation among 37 morphological traits and genome size (GS) data for 2016/17 (panel **a**) and 2017/18 data (panel **b**). Dark blue boxes indicate high values and dark red depict low values.

## 4. Discussion

To assess the diversity in a tarwi germplasm collection under Mediterranean conditions, phenotypic, genetic and genomic analyses were combined, studying morphological traits, ISSR markers and genome size. In general, the morphological traits used to evaluate accessions tested in the trials showed acceptable adaptability to the Mediterranean environment assuming that productivity projected is above 1.5 t/ha, achieved under rain-fed conditions unevenly distributed during the trial periods. Similar yields were previously reported in France and Spain [54]. However, the yields obtained suggest continuing breeding to achieve higher yields. The results showed also significant differences among accessions. While additional years of field trial results would certainly improve the robustness of conclusions, the results obtained are based on traits that presented coherent values between trials.

The knowledge of the correlations between different characteristics is fundamental because it allows the accomplishment of the indirect selection of the complex characteristics that are inherited quantitatively and influenced by genetic effects [43]. In this work we report positive and significant correlation between many variables. Characteristics such as total seed weight (TW) and total number of seeds (TNS) are very important and are directly related to characteristics of reproductive development. Accession LM268 presented higher values for TW and LM34, P20993 and LM18 for TNS. Therefore, these two features can work as criteria of selection in our collection for the breeding programme or to choose the most adapted. Talhinhas et al. [43] verified positive correlation between total weight with plant height, pod width, number of primary branches, proportion of seeds on the primary branches, total number of seeds and number of pods per primary branch in *L. angustifolius*. Georgieva and Kosev [55] also found positive correlation between thousand seed weight and plant height. Clements et al. [56] reported positive correlation between weight and plant height of 1000 seeds in *L. pilosus*. Heritability is a parameter widely used by breeders to genotype selection based on phenotypic expression [57]. Morphological traits exhibiting high values of heritability are chosen for the selection based on this parameter [43,58]. High heritability values enable the identification of important features to be selected for genetic breeding. Concerning our study, tarwi accessions can be selected based on the following traits: TW, SLMS, NPMS, NLMS, TNS, HUFF, PSMS, TSWMS, TSWPB, WSPPB, PWMS, SWBLR, TBL, ALPB, ADNL, SLPB, WSMS, WSPB, NPB, and NPPB. Similar results were noticed by Talhinhas [53] for SLMS, NLMS, HUFF, PSMS, TNS, and TW in *L. albus*, *L. angustifolius* and *L. luteus*. Our results also corroborate those by Georgieva and Kosev [55], who found high values of heritability for pod length and total number of seeds in *L. albus* and *L. luteus*.

In the present study several accessions stood out due to their superior performance in various traits. Along with accession LM268, accessions LM18, LM27, P20993, Potosi-ISA, PRT79 and I82 were the most efficient in converting vegetative growth to seed production, although lagging behind the performance of *L. albus*. Accession LM268 was the only tarwi accession to produce more yield on the primary branches than on the main stem, following the pattern of *L. albus* cultivars.

An important result worth highlighting is that most of the *L. mutabilis* accessions studied concentrate their production on the main stem. This characteristic is very important because it allows adaptability of tarwi to poor growth (soil and/or climatic) conditions. This characteristic may also prove useful to avoid the indeterminate growth habit of tarwi, particularly problematic in areas without summer drought. Breeding programmes should be directed for improving levels of production on the main stem and primary branches for good soil/climate conditions but with summer drought and focus on more determinate growth plants (those concentrating production on the main stem) both for marginal areas and for areas without summer drought. To the latter, accessions such as JKI-L295 and JKI-L210 stand out, as they produced over 80% of their yield on the main stem while attaining relatively high yields (ca. 10 g per plant).

The use of molecular markers in genetic diversity studies at the intra and inter specific levels proved useful in a wide range of species [59,60]. In this study we assessed the efficiency of ISSR markers for the characterisation of genetic diversity of *L. mutabilis* accessions. This technique is important because it allows to make a broad screening of a collection. SSR markers are not optimised yet for

*L. mutabilis* and the transfer of such markers from other Fabaceae to tarwi did not prove successful [61], leaving ISSRs as a valid tool for preliminary screening of germplasm collections. All six primers used in this investigation revealed a polymorphism of 30.55% for all 23 accessions. Bussell et al. [62] establish 20% as minimum of monomorphic band percentage for genetic diversity study and our study reveals 69.44% monomorphic bands. Similar results were reported by Chirinos-Arias et al. [37] assessing genetic variability among 30 accessions of *L. mutabilis* using eight ISSR markers, finding a total polymorphism of 58.82%. The high level of polymorphism obtained in our study is in accordance with those authors. The parameters PIC, EMR, MI and RP were used to evaluate the efficiency of ISSR primers. However, to the best of our knowledge, there are no studies on *L. mutabilis* assessing the effective multiplex ratio, polymorphic information content, marker index and resolving power. Results show high probability in detecting polymorphism PIC (0.72), for the primer $HVH(TG_7)$. The $AG_8YG$ primer stood out as presenting a high RP value (13.58), being more qualified to distinguish accessions. The highest value of EMR (2.25) was obtained with primer $AG_8YC$, revealing this to be most efficient. The primer $AG_8YC$ proved to be the most useful because it presented the highest value of MI (0.54). Several studies have been undertaken based on these techniques for selecting efficient ISSR primers in different species [63–67].

The 23 tarwi accessions were divided in five main genetic groups using cluster analysis by the UPGMA method (Figure 5). However, morphological characteristics such as stem and flower colour did not exhibit regular relationships in different clusters. The existence of several distinct groups that aggregate different stem and flower colours probably reflects few differences on the genetic constitution of the accessions. On the other hand, the distinct groups can reflect into distinct morphological characteristics and variations. Talhinhas [53] suggested that low intra-specific diversity in tarwi can be related to the fact that all the accessions originated from a limited number of landraces, reflecting the recent domestication genetic bottleneck effect that is estimated to have occurred no later than 2600 years before the present time in *L. mutabilis* [16]. Similar result was found by Chen et al. [68] in the research done on the 105 genotypes on *Vigna unguiculata*. In this work we verify that the genetic variability is not correlated with phenotypic variability, indicating the need for incorporation of more molecular markers. Similar results have been reported in other species. Previous studies performed by Galek et al. [14] also did not find a relation between genetic and morphological variability in accessions of *L. mutabilis*. In a study aiming to evaluate genetic diversity of *Nelumbo* using analyses of Randomly Amplified Polymorphic DNA (RAPD) and ISSR markers, Li et al. [69] found low correlation between molecular and morphological data. Talhinhas et al. [44] assessing genetic diversity in *Lupinus luteus* using ISSR and Amplified Fragment Length Polymorphism (AFLP) markers did not find any correlation between morphological and molecular data.

In this work we report the existence of significant differences in the intraspecific genome size (GS) variability in 23 accessions of *Lupinus mutabilis*. Our results reveal that the GS ranged from 1.94 pg/2C to 2.13 pg/2C. Naganowska et al. [70], also employing flow cytometry to analyse propidium iodide-stained nuclei, evaluated the nuclear DNA content variation in the genus *Lupinus* and found 1.90 pg/2C for *Lupinus mutabilis*, although a single accession was used in that study. To the best of our knowledge, our study is the first *L. mutabilis* genome size intra-specific analysis, depicting an overall average size of 2.05 pg (2001.2 Mbp). Several studies have reported intraspecific differences in genome size in various species such as *Glycine max*, *Linum austriacum* and *Zea mays* [33,71,72]. The intraspecific variation in genome size can result from repetitive/non-coding regions, hence increasing or decreasing in satellite DNA transposable elements and ribosomal genes [73]. There are studies pointing that transposable elements are largely responsible for notable differences in genome sizes. For instance, in maize, transposable elements are responsible for 85% of differences [74]. According to Petrov [75] these elements have potential of multiplicity of 0.1–1 Mbp in a single generation. The satellite DNA can also contribute greatly to genome size differences [76]. Meanwhile, Garrido-Ramos [74] refer that genomic content variation in plants which are affected by satellite DNA can range from 0.1% to 36%. Small variation of 3.5% in nuclear DNA have been associated with ribosomal genes [77]. The maximum

variation of nuclear DNA content obtained in the present research was 9.2%, a value much higher than the 2% maximum genome size variability reported for soybean [71] but smaller than the 36% variation reported for maize [72]. In light of this discussion, one may discard the possibility that differences in *L. mutabilis* genome size are caused by the transposable elements. Only a detailed study could unravel whether this variation is due to repeated sequence differences in satellite DNA or ribosomal genes.

Data on 37 morphological traits and genome size measurement were plotted and no correlation was observed. This is not a surprise, as similar results were also reported from other studies. For instance, Oney and Tabur [31] did not find correlation between genome size and morphological traits on the *Brachypodium distachyon* collected in different locations in Turkey. Realini et al. [72] observed weak association between genome size and morphological traits in maize. Recently Basak et al. [78] assessing the variation of morphological traits with the genome size in turnip found no correlation. This lack of association between morphological traits and genome size suggests that other factors are determinant on the control of such characteristics, reinforcing the view that genome size variations are mainly related to non-coding regions [79].

## 5. Conclusions

The agronomic performance of *L. mutabilis* in Portuguese conditions was good, assuming that the assay was conducted under rain-fed conditions. Our results highlight the accession LM268 with larger seeds and a total thousand seeds weight similar to *L. albus*, while also achieving the highest yield and being the only tarwi accession producing more on the primary branches than on the main stem. While high yields in lupins depend on the capacity of the plants to produce large amounts of pods and seeds on lateral branches, the indeterminate growth habit of tarwi can be undesirable, either in areas without summer drought or, on the contrary, in areas with limited growing periods where further vegetative growth may impair pod filling. To this end, JKI-L295 accession present high yield concentrated on the main stem, suggesting a semi-determinate development pattern. In either case, this accession is a key point for continued breeding. In fact, the present study has shown that tarwi is still behind white lupin in terms of its adaptability to Mediterranean conditions, namely concerning yield. The genetic diversity revealed in this study, however, prompts further breeding opportunities. Molecular marker and genome size analyses have revealed important levels of genetic/genomic diversity, which could not be related to phenotypic/morphologic diversity. This illustrates a scenario of recent domestication in the absence of a gene flow to wild relatives suggesting, however, that further exploitation of genetic diversity in this tarwi collection is possible and may provide additional sources of useful agronomic traits.

**Supplementary Materials:** The following are available online at http://www.mdpi.com/2073-4395/10/1/21/s1: Figure S1: Meteorological data collected at Tapada da Ajuda (Lisbon) during 2016/17 and 2017/18: (a) monthly average of daily average temperatures compared to the 30-year climatological normal values for Lisbon; (b) monthly rainfall compared to the 30-year climatological normal values for Lisbon; (c) soil water balance (including daily rainfall values), Table S1: Analysis of morphological traits by Kruskal Wallis test 2016/17 and 2017/18, Tables S2 and S3: Correlation matrix between morphological traits calculated for the 23 *Lupinus mutabilis* accessions under study (values greater than 85% are highlighted) for the experiment carried out in 2016/17 and 2017/18, respectively.

**Author Contributions:** Conceptualization, N.G. and J.N.-M.; Formal analysis, N.G., S.A. and P.T.; Funding acquisition, J.N.-M.; Investigation, N.G., S.A. and P.T.; Methodology, N.G., P.T. and J.N.-M.; Project administration, J.N.-M.; Supervision, P.T. and J.N.-M.; Writing—original draft, N.G.; Writing—review and editing, P.T. All authors have read and agreed to the published version of the manuscript.

**Funding:** This research was funded by the European Union (H2020/720726, LIBBIO project) and by Fundação para a Ciência e a Tecnologia, Portugal (UID/AGR/04129/2013, LEAF).

**Acknowledgments:** The authors acknowledge Julius Kühn-Institut for providing biological material, Instituto Português do Mar e da Atmosfera (IPMA) for meteorological data from the Tapada da Ajuda (Lisboa) weather station and Lusosem for the support provided in field trial conduction.

**Conflicts of Interest:** The authors declare no conflict of interest. The funders had no role in the design of the study; in the collection, analyses, or interpretation of data; in the writing of the manuscript; or in the decision to publish the results.

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
