# Peer review of "Genetic and Genomic Diversity in a Tarwi (Lupinus mutabilis Sweet) Germplasm Collection and Adaptability to Mediterranean Climate Conditions"

_agronomy, doi:10.3390/agronomy10010021_

Round 1
Reviewer 1 Report
Lupinus mutabilis is an alternative rich protein and oil source to soybean and potential crop in Europe and globally. However, there is limited knowledge of its intraspecific variability using morphological and genomic tools. This manuscript employed morphological traits, ISSR markers, and genome size to assess genetic and genomic diversity in 23 L. mutabilis accessions. An only shortcoming is a limited number of accessions (23) used in this study which is small germplasm.
A very good standard of writing, in terms of logic and correctness, except typos in the text and language issues. Methods are clear, concise and accurate. The results are clearly presented. The discussion is very focused in line with the research question.
Author Response
English language was thoroughly revised.
Reviewer 2 Report
There are some spelling and grammatical errors throughout the manuscript that can be corrected after the proofreading service done by a qualified native speaker of English.
Authors, please explain why you did not use other molecular markers in addition to ISSR, and why an additional year of field trials was not included in this study. The importance of these two issues must be addressed and discussed in the manuscript.
Fig 6 should be replaced as it is not of good quality. I suggest using a photo of a GT8YC primer gel instead of the current one.
For which year were the samples used for the ISSR analysis? 2016/17, 2017/18 or bulk of both years?
Author Response
There are some spelling and grammatical errors throughout the manuscript that can be corrected after the proofreading service done by a qualified native speaker of English.
Reply: English language was thoroughly revised.
Authors, please explain why you did not use other molecular markers in addition to ISSR, and why an additional year of field trials was not included in this study. The importance of these two issues must be addressed and discussed in the manuscript.
Reply: Thank you for the comment. We agree that other markers can be more informative. We have changed the text in introduction and discussion to stress that: ISSRs were employed in this study for initial screening of a germplasm collection; more informative markers, such as SSRs, are not optimized yet for L. mutabilis. Other molecular traits, such as nucleotide sequence of taxonomically informative genes, are mostly invariable among accessions. We also agree that additional years of field trial results would certainly improve the robustness of conclusions. The results present and the conclusions derived are based on traits that were coherent between years. The discussion was changed to stress this.
Fig 6 should be replaced as it is not of good quality. I suggest using a photo of a GT8YC primer gel instead of the current one.
Reply: A better version of the electrophoresis depicted in Figure 6 is now in the manuscript.
For which year were the samples used for the ISSR analysis? 2016/17, 2017/18 or bulk of both years?
Reply: Plants for molecular analyses (ISSRs and genome size) were specifically sown for these purposes, using the same seed stocks from which seeds were picked for sowing each of the field trails.